# Union-Retire for Connected Components Analysis on FPGA

**DOI:** 10.3390/jimaging8040089

**Published:** 2022-03-24

**Authors:** Donald G. Bailey, Michael J. Klaiber

**Affiliations:** 1Centre for Research in Image and Signal Processing, Massey University, Palmerston North 4442, New Zealand; 2Computer Science, Baden-Wuerttemberg Cooperative State University (DHBW), 70178 Stuttgart, Germany; michael.klaiber@lehre.dhbw-stuttgart.de

**Keywords:** union-find, connected components, feature extraction, pipelined, FPGA

## Abstract

The Union-Retire CCA (UR-CCA) algorithm started a new paradigm for connected components analysis. Instead of using directed tree structures, UR-CCA focuses on connectivity. This algorithmic change leads to a reduction in required memory, with no end-of-row processing overhead. In this paper we describe a hardware architecture based on UR-CCA and its realisation on an FPGA. The memory bandwidth and pipelining challenges of hardware UR-CCA are analysed and resolved. It is shown that up to 36% of memory resources can be saved using the proposed architecture. This translates directly to a smaller device for an FPGA implementation.

## 1. Introduction

In embedded vision applications, three factors are often of importance: real-time processing (large amounts of image data cannot be stored for later offline processing), processing latency (especially important for stability when vision is part of a control system) and power (many embedded applications have a limited power budget). Where these requirements cannot be met by conventional software systems, many have turned to field programmable gate arrays (FPGAs) for their implementation. The parallel nature of hardware design for an FPGA can often accelerate a design, enabling operation at a lower clock frequency (reducing power requirements) and significantly reduce the latency because images can be processed directly as they are streamed from a camera without first having to save them in memory (as is required by software). However, these benefits come at the cost of the increased design effort that is required for a parallel hardware design, as is demonstrated within this paper.

Many image analysis applications use some form of connected components analysis (CCA) to measure features of objects within an image. First, preprocessing operations are used to segment the objects of interest within the image, resulting in a binary image where each object is comprised of a set of connected pixels. Each connected component can then be extracted, enabling a vector of features of the object to be calculated. These feature vectors can then be used to eliminate false objects (or noise) or otherwise classify each object into one of several classes, and provide key data about each object.

Typically, connected components are identified and labelled using some form of Union-Find algorithm [1]. Labels are initially assigned to pixels based on connectivity. When two different labels are detected as belonging to a single object (for example when two sub-components are initially assigned different labels, and these sub-components are later detected to be adjacent), a Union operation combines the sets of equivalent labels. Since each object can be represented by sets of labels, a Find operation is required to determine the representative label for a component. A connected components labelling (CCL) algorithm therefore requires another pass through the image to replace the set of labels associated with a component by the single representative label.

For real-time processing on an FPGA, multiple passes through the image require the image to be buffered (usually in external memory) and this significantly increases the latency. Therefore, single-pass CCA algorithms were developed that calculate the feature vector of each component during the labelling process [2]. Data are accumulated incrementally as each pixel is added to the component. A Union operation requires combining the feature vectors of the sub-components, with the resultant feature vector associated with the representative label (identified using a Find operation) [1]. This combination of feature vectors by a Union requires that the components of the feature vector are able to be accumulated associatively [2].

Union-Find algorithms focus on labelling an image. However, for CCA, the labels are only an intermediate step, and the direct output is the feature vector for each connected component rather than a labelled image. The real focus needs to be on the connectivity between sub-components, rather than their actual labels, and this has led to the recently proposed Union-Retire algorithm [3]. Each run of consecutive object pixels on an image row is assigned a label. A Union operation is still required to link runs when they overlap between rows, but the Find operation is no longer required. Instead, a Retire operation is used when a run is no longer accessible (during the raster scan of the image). The Retire operation has two functions. First, it removes the label from the set, while ensuring that the remaining members remain connected. Second, it passes the partial feature vector to one of the remaining members of the set to ensure that its data are included in the final feature vector.

This paper explores a pipelined, streamed implementation of the Union-Retire algorithm on an FPGA, and analyses the requirements to make it work. In many low-cost FPGAs, on-chip RAM (random access memory) is often the most precious resource when implementing image processing algorithms. Therefore, within this paper, we have focused on the ability of the Union-Retire algorithm to reduce memory requirements when implementing CCA.

## 2. Related Work

Early CCL algorithms [4,5] required two or more passes to assign a distinct label to each component of a binary image. The need to process high-resolution images and image streams with an increasing frame rate, however, made such algorithms expensive with regards to computer and memory resources which required optimisations for hardware realisation on FPGAs. Here, mainly FPGA and hardware-centric publications are discussed. For further reference, the review by Bailey [2] provides a detailed overview of the history of single-pass CCL and CCA algorithms.

Bailey and Johnston [6] shifted the focus from creating a labelled image to extracting and accumulating feature vectors while scanning the input image. Consequently, the name of the described algorithm was shifted from connected components labelling (CCL) to connected components analysis (CCA). The ability to process an input image with a single pass also significantly reduced the required memory resources, which was especially important for processing on FPGAs with limited memory [7]. This idea resulted in a number of optimisations to reduce memory requirements, e.g., aggressive relabelling [7], double lookup [8], and deferred label assignment [9].

These algorithms still required mechanisms to resolve chained dependencies of labels which are a result of the raster-scan processing. In [6] the resolution period was at the end of each image row (during horizontal blanking). This was eliminated by some of the more recent algorithms [10,11,12].

Algorithms that process one pixel per clock cycle have an inherent sequential nature of the labelling and merging process which make them difficult to accelerate through parallelisation. A number of parallelisation methods have been proposed. Kumar et al. [13] parallelised the labelling of CCL by processing horizontal slices of the input image. These are combined afterwards in a sequential process. Klaiber et al. [14] extract feature vectors from vertical slices of the image in parallel from which results are combined in a scalable coalescing unit [15]. Kowalczyk et al. [16] demonstrate an implementation of processing up to four pixels in parallel with neighbour-pixel parallelisation.

Lacassagne and Zavidovique [17] identified memory accesses and conditional statements to be the key issue slowing down CCL on RISC (reduced operation set computing) processors. In [18] this was identified to require the fewest processing cycles per pixel when carried out on a general-purpose processor. Newer versions of this algorithm also consider vector processing [19]. Even though [17] has not been directly realised as hardware architecture for CCL/CCA, this algorithm considers many hardware aspects, such as memory hierarchy and data path, which provided inspiration for FPGA implementations.

All of the aforementioned algorithms use a variant of Union-Find as the set-merge algorithm to combine sets of equivalent labels when merging regions, and to find the representative label of a component. Consequently, the focus of many algorithms is on the labelling, rather than the connectivity, which is not surprising given that they are derived from algorithms where the key aim is to produce a labelled image.

Tang et al. [20] is the first to move away from this, with each run of consecutive object pixels assigned a unique segment identifier, with the focus more on the linking of segments to convey connectivity. Based on this, an alternative approach to Union-Find is proposed called Union-Retire [3]. Like Tang et al., it deterministically assigns an index to each run, and uses a linked-list structure to represent mergers. However, rather than having a linear structure, with the head as the representative label, a more general graph structure is proposed to represent the connections between runs.

## 3. Union-Retire Algorithm

The focus of this paper is the realisation of a hardware architecture of the Union-Retire algorithm for CCA (UR-CCA) presented in [3]. Therefore, this section provides a brief overview of the UR-CCA algorithm. A more detailed description and examples can be found in [3].

UR-CCA shifts the paradigm of CCA algorithms from Union-Find-based operations on a forest data structure to using a more general graph. The basic principle is to represent each run of consecutive object pixels as a node within the graph, where each connected subgraph corresponds to a single connected component. When a run is no longer accessible, a Retire operation removes the corresponding node from the graph, simplifying the graph’s structure. Throughout this process, the focus is on maintaining the overall connectivity within the component between the constituent runs.

Algorithm 1 shows the operations for Union-Retire dependent on the content of the current 2×2 window and the previously processed pixels in the image represented by the state of the graph. To simplify the description, the window patterns are shown graphically, where background pixels are shown in white (
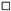
), object pixels in black (
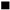
), and pixels that can be either object or background are shown in grey (
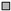
).

Each node in the graph is assigned an index. All nodes in the graph have a maximum of two outgoing arcs, represented by a link table (an array) LT[] with fields for the two links $LT\left[\;\right].L_{1}$ and $LT\left[\;\right].L_{2}$ (a shorthand notation is later used to represent the most recent link, i.e., LT[].L refers to $LT\left[\;\right].L_{2}$ when $LT\left[\;\right].L_{2}\neq 0$, otherwise $LT\left[\;\right].L_{1}$). A feature vector is associated with each node and stored in a data table DT[]. The image is scanned in raster fashion, using a 2×2 local window to determine connectivity. Two counters, *P* and *C*, contain the most recent indices on the previous and current rows, respectively. These are updated (incremented) when a new object pixel is found in the corresponding row (previous row: 
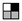
, line 5; current row: 
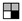
, line 8).

The feature vector (stored in DT[]) for the node on the current row is initialised at the start of a run (
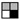
, line 8), and updated for each object pixel within a run on the current row (
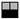
, line 12), where ⊕ is the operation which combines two feature vectors.

When a run in the previous row is connected to a run in the current row, a Union operation is invoked, which adds a link between the corresponding nodes to connect the two sub-components. This is detected on the end of a run on either row when there is an object pixel on the other row (
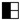

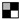

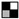
, line 15). This link is always added from P→C to facilitate the Retire operation. If node *P* already has a link then the link is added to its successor so that several successive nodes are linked in a chain (see lines 19 and 22). If the successor node already has two links then the link is propagated to the successor’s most recent successor (line 44).
**Algorithm 1** Union-Retire CCA algorithm (adapted with permission from ref. [3]. Copyright 2021 Springer.)**Input:** Binary image *I* of width *W* and height *H*
**Output:** A feature vector for each connected component in *I*
1:P≔C≔0                                ▹ Initialise run indices2:**for** y≔0 **to** 
*H* 
**do**3:   **for** x≔0 **to** *W* **do**                        ▹ Raster scan through the image4:     Window ≔ I[x−1:x;y−1:y]                      ▹ Form 2×2 window5:     **if** Window = 
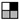
 **then**                          ▹ Start of run in previous row6:        P+=17:     **end if**8:     **if** Window = 
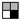

**then**                         ▹ Start of run in current row9:        C+=110:        $LT\left[C\right].L_{1}≔LT\left[C\right].L_{2}≔0$                        ▹ No links yet11:        DT[C]≔F(x,y)                         ▹ Feature vector for pixel12:     **else if** Window = 
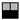

**then**                       ▹ Continuing current run13:        DT[C]≔DT[C]⊕F(x,y)              ▹ Accumulate feature vector for each pixel14:     **end if**15:     **if** Window **in** {
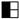

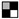

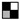
} **then**                     ▹ Union operation, link nodes16:        **if** $LT\left[P\right].L_{1}=0$
**then**                             ▹ No links17:          $LT\left[P\right].L_{1}≔C$18:        **else if** $LT\left[P\right].L_{2}=0$
**and**
$LT\left[P\right].L_{1}\neq C$
**then**                ▹ Only one link19:          Link($LT\left[P\right].L_{1}\to C$)20:          $LT\left[P\right].L_{1}≔C$21:        **else if** $LT\left[P\right].L_{1}\neq C$
**and**
$LT\left[P\right].L_{2}\neq C$
**then**                  ▹ Two links22:          Link($LT\left[P\right].L_{2}\to C$)23:          $LT\left[P\right].L_{2}≔C$24:        **end if**25:     **end if**26:     **if** Window = 
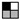

**then**                       ▹ Retire operation, unlink node27:        **if** $LT\left[P\right].L_{1}=0$
**then**                                ▹ No links28:          **Output:** DT[P]                                ▹ Object completed29:        **else if** $LT\left[P\right].L_{2}=0$
**then**                            ▹ Only one link30:          $DT\left[LT\left[P\right].L_{1}\right]≔DT\left[LT\left[P\right].L_{1}\right]⊕DT\left[P\right]$            ▹ Accumulate feature data31:        **else**                                              ▹ Two links32:          $DT\left[LT\left[P\right].L_{2}\right]≔DT\left[LT\left[P\right].L_{2}\right]⊕DT\left[P\right]$            ▹ Accumulate feature data33:          Link($LT\left[P\right].L_{1}\to LT\left[P\right].L_{2}$)                      ▹ Maintain linkages34:        **end if**35:     **end if**36:   **end for**37:**end for**38:**procedure**Link(X→Y)                               ▹ Link two nodes39:   **if** $LT\left[X\right].L_{1}=0$
**then**                                 ▹ No links40:     $LT\left[X\right].L_{1}≔Y$                                     ▹ Add link41:   **else if** $LT\left[X\right].L_{2}=0$
**and**
$LT\left[X\right].L_{1}\neq Y$
**then**                 ▹ Only one link42:     $LT\left[X\right].L_{2}≔Y$                                 ▹ Add second link43:   **else if** $LT\left[X\right].L_{1}\neq Y$
**and**
$LT\left[X\right].L_{2}\neq Y$
**then**                  ▹ Two links44:     Link($LT\left[X\right].L_{2}\to Y$)                                ▹ Pass link on45:   **end if**46:**end procedure**

At the end of a run on the previous row (
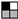
, line 26), the associated index is never accessed again. Therefore, a Retire operation is performed, removing the corresponding node from the graph. The feature vector associated with the retired node is accumulated into a linked node (⊕ on lines 30 and 32). If there are no linked nodes then the object is detected as completed, and the feature vector of the connected component is output. If there are two linked nodes then it is necessary to insert a link between the nodes to maintain connectivity within the graph as the node is removed.

To simplify the description, Algorithm 1 omits end-of-line corner cases, and therefore expects a blank pixel at the end of each line. The hardware realisation explicitly handles these cases without the need to add an extra column of blank pixels.

## 4. Hardware Implementation

When implementing the UR-CCA algorithm on an FPGA, we are working with the following constraints:The input image is assumed to be streamed in a raster manner with a throughput of one pixel per clock cycle. This will require many of the operations to be pipelined, especially those that update data structures in memory.The hardware realisation must be able to work on a continuous stream, i.e., one without any blanking at the end of the line (EOL) or frame (EOF). This implies that additional blank pixels cannot be inserted at the end of line and frame to facilitate transition from one image row to the next, or from one frame to the next.To minimise latency, it is desired that the feature vectors for each connected component be output as soon as the end of the component is detected.

As indicated in Algorithm 1, the link and data processing are loosely coupled, simplifying the hardware architecture. The architecture for UR-CCA comprises three main blocks as identified in Figure 1:Forming the 2×2 window from the incoming binary pixel stream, and assigning each run of pixels a corresponding node index.Link processing identifies linkages between runs on the current row and runs on the previous row, and maintains the graph structure, which represents the connectivity as each pixel is processed.Data processing builds the feature vector associated with each connected component, and outputs this when the component is completed.

### 4.1. Forming the 2×2 Window

The input pixel stream is a continuous sequence of binary pixel data, following a standard row-wise raster scan, as illustrated in Figure 2. The absence of blanking between rows and frames requires two control signals, one to indicate the last pixel in each row (EOL), and the other to indicate the last pixel in a frame (EOF).

The 2×2 window processing is straight forward to implement, with the detailed architecture represented in Figure 3. A 2×2 array of one-bit registers contains the neighbourhood pattern associated with processing the current pixel. The row buffer delays the incoming pixels by one row, making them available when processing the next row.

Each run of consecutive object pixels is represented by a node in the connectivity graph. Since the node indices are allocated sequentially, a run counter suffices for labelling the runs on the current row (the *C*-counter). This increments the node index at the start of a new run (an object pixel following a background pixel (
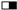
), or an object pixel in the first pixel of a row). Once a run leaves the previous row within the window, it is never encountered again. This enables the indices associated with the nodes to be recycled, reducing the size of the link and data tables. The maximum number of indices required at any one time is based on the width of the image [3], and is given by $\lfloor \frac{W}{2}\rfloor +2$. Therefore, the counter goes from 1 for the first run, up to $\lfloor \frac{W}{2}\rfloor +2$, after which it resets to 1 again. Note that index 0 is reserved to indicate empty links within the link table.

The indices are also deterministic (they do not depend on any linking operations), so the runs on the previous row can also be directly labelled using a similar counter (the *P*-counter). This avoids the need for the row buffer and window to contain the actual labels (as most previous label-based methods require). A single counter for each window row is sufficient (rather than labelling each window pixel) because there can be at most one active index on each row of the window at a time. The exception is the transition from one row to the next, where the last pixel of the previous row, and the first pixel of the current row (two different indices) are both within the window; the handling of this EOL case by the link and data processing will be discussed in the following sections.

The outputs of this module are the node indices from the two run counters and the 2×2 window pattern, which are used to control the link and data processing modules.

### 4.2. Link Processing

The UR-CCA algorithm, as originally formulated, assumes that each pixel is processed completely before considering the next pixel in the incoming pixel stream. However, when using a memory-based data structure to represent the graph, a hardware system is unable to realise the required operations within a single clock cycle. Adding a link between two nodes (X→Y) requires two memory operations: the first to read LT[X] to determine the current outgoing links associated with node *X*, and a second to write the updated links back to memory. Therefore, on an FPGA with synchronous memory, adding a link will need to be pipelined over at least two memory clock cycles. Delaying the update as a result of pipelining may potentially create read-before-write data hazards (reading a link from memory before it has been properly updated).

Since a link is always from an earlier index to a later index, node *P* (associated with the run on the previous row) is always updated whenever a link is added. In addition, a chain link is also added (so that each node may have at most two outgoing links [3]). This creates a potential bandwidth issue with multiple links having to be updated each clock cycle. To reduce this bandwidth, the link table entry associated with the node on the previous row, LT[P], is cached within a register, $L_{P}$, enabling it to be accessed immediately, and updated within a single clock cycle.

In terms of pixel patterns within the 2×2 window, there are four key operations on the link table:N-operation: 
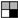
, at the start of a new run on the current row. This requires the link table entry for the new node to be initialised by clearing the links to zero (since node indices are recycled, and the link table may contain old links), LT[C]≔{0,0} (a write operation).P-operation: 
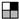
, at the start of a run on the previous row. The link table entry for the corresponding node is read into the cache to facilitate adding links, $L_{P}≔LT\left[P\right]$ (a read operation).L-operation: one of 
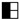

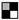

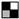
, at the end of a run on the current or previous row, while there in an object pixel in the other row. A Union adds a link from the node on the previous row to the node on the current row, P→C. There are two variations to this: if the cache is empty ($L_{P}=\left\{0,0\right\}$), or the cache contains a link.−If the cache is empty, an F-operation adds the first link to the cache, $L_{P}≔\left\{C,0\right\}$ (no memory operations).−Otherwise, a chained link is added to the graph from the most recent link, $L_{P}.L\to C$ (this requires a read and a write memory operation), and the link is replaced, $L_{P}.L≔C$.R-operation: 
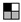
, at the end of a run on the previous row, a Retire removes node *P* from the graph. If the cache has fewer than two links ($L_{P}.L_{2}=0$) then no operation is required because removing the node does not affect the connectivity. Otherwise, a link must be added between the two connected nodes to maintain connectivity of the object graph when node *P* is removed, $L_{P}.L_{1}\to L_{P}.L_{2}$ (a read and a write).

#### 4.2.1. Link Table Memory Bandwidth

In terms of these operations, the worst-case image pattern is a checkerboard pattern, shown in Figure 4. From this pattern, the following memory accesses are implied:An N-operation every second pixel requires a write in every second cycle;A P-operation every second pixel requires a read in every second cycle;An L-operation with every pixel requires a read followed by a write in every clock cycle;An R-operation every second pixel requires, in the worst case, a read followed by a write in every second cycle.

Based on this worst case, two reads and two writes are required for every pixel, whereas the dual-port RAMs available on FPGAs only allow two accesses total per clock cycle.

However, a more careful analysis reveals that many of these accesses are redundant. This redundancy can be exploited by implementing the operations using the following rules:1.Loading the cache for a P-operation must be performed immediately. This ensures that the links are available for subsequent operations.2.An F-operation only updates the cache; as this is the first link, a chained link is not required.3.If a P- and L-operation occur in the same cycle (data pattern: 
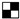
), then adding the link must be deferred until the following clock cycle because the links for the previous row have not yet been loaded into the cache.4.If an L-operation is deferred and this is followed immediately by another L-operation (
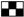
) then there will be a chain from the first link to the second, C−1→C. Since links are always from an earlier node to a later node, LT[C−1] must be empty, therefore the read operation for the chained link is not required, and the link can be written directly to the table, LT[C−1]≔{C,0}. (This saves 1 read operation.)5.If as a result of adding a link, LT[C] is read in the cycle immediately before a P-operation, then the cache read is redundant (as the index has just been read). The subsequent write-back of the updated link entry is also not required, as the update can be made directly on the cache. (This saves 1 read and 1 write operation.)6.If when adding a link to a node, the destination node is already linked, then the write is not necessary. (This saves 1 write operation if the write is to memory.)7.If, when adding a link (X→Y), node *X* already has 2 outgoing links, it is necessary to pass the link on, $LT\left[X\right].L_{2}\to Y$. (This requires 1 extra memory read, but does not occur very frequently.)8.If an L- and R-operation occur in the same cycle (
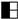
) then the cache is updated to reflect the link, the retirement link is queued first, ($L_{P}.L_{1}\to C$) with the chained link queued second ($L_{P}.L_{2}\to C$). $L_{P}.L_{1}$ occurs earlier than $L_{P}.L_{2}$, so this reduces potential data hazards.9.Initialising the link entry for an N-operation has the lowest priority, as it cannot be used for at least two clock cycles.

The result of applying these rules to the worst-case image of Figure 4 is shown in Figure 5. Runs (nodes) are assigned an index from ① to ⑦, after which retired indices are recycled, starting with ① again.

Processing row 0 consists of a series of N-operations to initialise the link table for the new nodes.

In row 1, the P-operation loads the outgoing links for each node in the previous row from the link table into the cache (for example, cycles 12 and 13). The link in cycle 13 is added directly to the cache. In cycle 14, the link from ② to ⑦ cannot be processed immediately, because ② has not yet been loaded into the cache. When the link entry for ② is available in cycle 15, the links from ② to both ⑦ and ① are processed in parallel. The cache is adjusted to contain the link to ①, and the chained link from ⑦→① can be written directly to memory. Note that the new link entry for ① is queued for writing in the following clock cycle. Similar processing is repeated for the remainder of the row.

In row 2, a similar process is used to link from ⑦ to ⑤ and ⑥ in cycles 24 and 25. However, this time ⑦ already has a link (to ①). This requires two chained links to be added: from ①→⑥, and ⑤→⑥. The link from ⑤→⑥ can be written directly, but the link from ①→⑥ requires first reading the link table for index ① (in cycle 25). In cycle 26, it is necessary to load the entry for ① into the cache; however, ① has just been read so this has three implications. First, it is not necessary to write the updated ① (from the chain) to memory, the update can be made directly to the cache; second, since the entry for the index has just been loaded, another read for the P-operation is unnecessary; and third, the target of this L-operation is the same as the target of the previous link, so the link is unnecessary (it is already present in the cache). Then in cycle 27, the link from ① to ⑦ results in the chain ⑥→⑦. When ① is retired, it has two outgoing links, so it is necessary to add a link from ②→⑦ to maintain connectivity. The read of ② in cycle 27 is similarly in advance of loading the cache. This pattern repeats in the following cycles in row 2, with a similar pattern continuing on subsequent rows.


These optimisations introduced have reduced the required link table bandwidth for this example to one read every second cycle, and one write every cycle. This implies that a single dual-port RAM is sufficient (with a queue for read and write accesses that cannot be performed immediately).

#### 4.2.2. EOL Queuing

The next example (shown in Figure 6) has a wider range of patterns, with the timing of link table accesses illustrated in Figure 7. In particular, issues associated with the transition from one row to the next are identified and discussed.

Rows 0 and 1 are similar to before. Chaining can be clearly seen in cycle 21. Before the L-operation, the cache contains a link to ③, which is replaced by ④. The chained link from ③→④ is added to memory to maintain connectivity; ③ is read in cycle 21, and updated in cycle 22. At the end of row 1, another L-operation results in another chain, and the initialisation of ⑤ from the N-operation is queued.

However, at the start of row 2 (cycle 23 in Figure 7), run ⑤ from the previous row is terminated by the row end, but is contiguous in the window with the new run, ⑥. This results in an N-operation which must be queued. In addition, the write phase for the previous chain, ④→⑤ is also due. The previously queued initialisation of ⑤ is executed, with the other two writes queued, requiring the write queue to be at least two entries long.

This situation does not occur in normal processing within a row because each run is terminated by a background pixel so that successive N-operations are at least two clock cycles apart. In this case though, the run ⑤ is terminated by the EOL signal, requiring N-operations in consecutive clock cycles.

A second EOL issue can be seen in cycles 33 and 34. Normally the link ⑤→③ would result in an L-operation in cycle 34 (like a similar L-operation in cycle 29). However, since this is at the start of the next row, both the *P*-counter and *C*-counter have already been incremented because of the start of runs on the new row. This is solved by detecting the end of both runs ⑤ and ③ (from the *EOL* signal), and initiating the L-operation in that cycle.

The other feature of note in this example is the application of rule 7 in cycle 36 (and again in cycle 40). When adding the link ⑦→④, it is discovered that ⑦ already has two outgoing links. Therefore the link is passed on to its second linked node, ②→④, with the node being written in cycle 37.

#### 4.2.3. EOF Processing

It is also important to manage the transition from one frame to the next to prevent runs on the last row of a frame linking to those on the first row of the next frame. The simplest approach to handle this is to disable the L-operations within the link processor when processing the first row of a frame. This implicitly places a blank row below the last row and above the first row of a frame, without requiring additional clock cycles. It also means that additional processing (and logic) is not required on the last line for detecting completed components. Any objects which extend onto the last image row will be detected as completed while processing the first row of the next frame using the existing logic.

However, to maintain link and data integrity, the run counter, *C*, cannot be reset to zero at the start of a new frame as this could result in overwriting links and data for unfinished connected components in the previous frame. This is managed by simply continuing the count sequence from one frame to the next.

#### 4.2.4. Link Table Architecture

The link table processing is implemented with the links stored in a dual-port on-chip memory as illustrated in Figure 8, with separate ports used for read and write access to the link table.

Read accesses are triggered either by:A P-operation to load the cache, with the address coming from the *P*-counter;When adding a link into the memory, either when adding a chain from an L-operation, or from an R-operation, in both cases, the address comes from the cache;When adding a link, and the link table already has two entries, the next access is to one of the links just read.

Note that the synchronous nature of the block memories require a dedicated register for $Data_{R}$.When adding a link, it is also necessary to register both the address ($Addr_{R}$) and the index being added (WVal) to enable the write to be pipelined into the following cycle. If two reads are required in any cycle, one is stored in the read queue and delayed until the following cycle (the read queue must also store the index being written into the memory). A single entry for the read queue has been found to be sufficient. WVal also holds the index *C* when a link is deferred (as a result of rule 3) in case it changes in the following clock cycle.

Link entries are written to the write port under two conditions:Clearing prior use of the index by an N-operation;Writing the updated outgoing connections for a node when a link is added.

The ‘update’ block in Figure 8 inserts the link being added into the appropriate field within the link table entry. The write queue manages the situation when multiple writes are required in a cycle. As demonstrated earlier, up to two pending writes may need to be queued.

The cache register, $L_{P}$, holds the current entry from LT[P]. To enable the value read from a P-operation to be used immediately, the actual cache value is the the output of the multiplexer between the cache register and the output of the link table. The ‘replace’ block replaces the most recent index with the target of a link. The entry is replaced, rather than inserted, because of the chaining mechanism. $L_{P}.L$ is also output to the data processing module to enable the feature data to be merged appropriately when node *P* is retired.

### 4.3. Data Processing

The role of the data processing block is to accumulate the feature vector for each connected component in the image. Conceptually, the processing performed is quite straight-forward:Data are accumulated for the run of pixels on the current row ($F_{C}\;⊕=F\left(x,y\right)$, where F(x,y) is the feature vector for the current pixel) and saved into a data table at the end of the current run ($DT\left[C\right]≔F_{C}$).When a node is retired (at the end of the run on the previous row), its associated feature datum, $F_{P}$, is merged into the data table entry of its most recent outgoing link. Let the merge link be M≔LT[P].L, then the data merging is $DT\left[M\right]⊕=F_{P}$.If there are no outgoing links, the connected component is completed, and the feature vector, $F_{P}$ is streamed out.

The memory accesses to the data table are therefore:At the end of the run on the current row (window pattern 
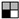
), the accumulated feature vector for the row is written to the data table: $DT\left[C\right]≔F_{C}$.At the start of the run on the previous row (window pattern 
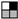
), the feature vector $F_{P}$ is read from memory, $F_{P}≔DT\left[P\right]$, in preparation for merging with data from another run when the node is retired.At the end of the run on the previous row (window pattern 
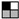
) when the node is retired, it is necessary to merge the feature vector into the feature vector of an outgoing link.−If there is an object pixel on the current row (
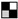

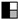
), then this will have the index of the most recent link, and the data can be merged directly into the current row accumulator ($F_{C}⊕=F_{P}$).−Otherwise (
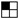
) the merge index comes from the link processor cache, $M≔L_{P}.L$. This requires reading the data table entry for the link, accumulating the feature vector, and writing the result back to memory: $DT\left[M\right]⊕=F_{P}$. The data table read takes one clock cycle ($F_{M}≔DT\left[M\right]$) with the accumulated feature vector written back in the following cycle ($DT\left[M\right]≔F_{M}⊕F_{P}$).

#### 4.3.1. Data Hazards

Under normal operation (not considering EOL processing), the window pattern 
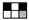
 can potentially create a data hazard. The node associated with the first run, $P_{1}$ is retired when the background pixel is encountered. The data table for the outgoing link, $DT[M_{1}]$ is read to accumulate the feature vector into. The combined feature vector is written back to the data table in the following clock cycle. However, in this cycle, the run $P_{2}$ is encountered with $DT[P_{2}]$ being loaded from memory. If $P_{2}=M_{1}$ then this results in a read-before-write hazard because $DT[M_{1}]$ is not updated in memory until the end of that cycle. Such linking between consecutive runs is actually quite common as a result of chaining. This can easily be handled by data forwarding, skipping both the write of $DT[M_{1}]$ and the read of $DT[P_{2}]$, and the accumulated data directly forwarded to $F_{P}$.

Since synchronous memory reads must be loaded into a dedicated memory register, any data forwarding must be to a separate register with a multiplexer to select between the memory register and the data forward register.

#### 4.3.2. EOL Processing

Under normal operation there can be at most one read and one write of the data table in any clock cycle. This suggests that a simple dual-port memory provides sufficient access when storing the data table. Indeed, this will be the case if the image is padded with one extra column of blank pixels at the end of each row. However, as seen with link processing, the constraint of not introducing an extra column of blank pixels means that EOL processing can disrupt the regular access patterns. Here, we identify three additional data hazards and three data table access issues that EOL processing must handle.

Consider the following window pattern: 
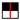
 where the red line indicates the transition between one row of image pixels and the following row. The EOL ends run $P_{1}$ on the previous row, resulting in retirement of the associated node, and starts a new run (with a new index, $P_{2}$). The two pixels in the window are not directly connected; they are separated by the EOL. The retirement requires reading the linked entry, $DT[M_{1}]$ for passing the feature data on to, and the new run also requires a read of $DT[P_{2}]$ to load the feature data for that node. This pattern therefore requires two simultaneous data table reads.

There is also a potential data hazard if the first node links directly to the second ($M_{1}=P_{2}$). The data accumulated from the retired node will not be saved before the entry is read. However, in this case, both read addresses will be the same, and the data from the retired node can be accumulated directly into $F_{P}$.

Another potential hazard can occur in the following clock cycle with 
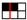
. In this case, if the two nodes both link directly to a common third node ($M_{1}=M_{2}$), the data for that first retirement will be written to $DT[M_{1}]$ in the same cycle as the data is read for accumulating the second retirement, $DT[M_{2}$]. In this case, the second read can be skipped, with the data for the second retirement accumulated directly.

A third pattern of interest that gives a data hazard is 
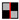
 when the indices for the two runs are the same (P=C; this will be the case if the run occupies the full width of the image). The read (for the run on the previous row) is to the same address as the write (for saving the feature vector for the current run). Again this can be managed through data-forwarding, skipping both the read and the write.

Finally, the pixel pattern 
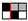
 requires two simultaneous writes. The first is the pipelined write for the retirement of the node on the previous row (DT[M]), and the second is for saving the feature vector for the node on the current row (DT[C]). The previous case of two simultaneous reads and this case of two writes mean that the data table must use true dual-port memory.

Even more serious is the particular pattern 
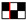
, which requires a read in addition to the two writes. In this case, the write-back of DT[M] is delayed until the following clock cycle.

#### 4.3.3. EOF Processing

As with the link table processing, the detection of feature vector mergers from the previous row to the current row is disabled when processing the first row of an image. This prevents the data from the last row of one frame being accumulated into the data for the first row of the next frame. The regular data merging process is then able to correctly merge data for object on the last row of an image, including detecting object completion.

Similar to the previous pattern, 
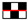
 (where the horizontal red line represents the boundary between successive frames) also requires a read and two writes. As before, the write-back of DT[M] can be delayed until the following clock cycle.

This pattern can also generate a data hazard, in the same way as that described earlier in Section 4.3.1. Again this is managed by data forwarding, with the associated read and write both skipped.

#### 4.3.4. Data Table Architecture

In all of the cases described here, a maximum of two data table memory accesses are required in any cycle. Therefore the data table can be implemented using a true dual-port memory block as demonstrated in Figure 9, with the reads and writes allocated depending on the particular window pattern. Referring to Figure 9, port 1 is used for reading $F_{P}≔DT\left[P\right]$ and writing the merged data back to DT[M]. Port 2 is used for reading $F_{M}≔DT\left[M\right]$ and writing $DT\left[C\right]≔F_{C}$. It is also used for writing DT[M] when port 1 is busy reading $F_{P}$. *C* needs to be registered (as $C_{W}$) in case it changes before writing $F_{C}$ to memory (for example at the end of a row). *M* is also registered (as $M_{W}$) for writing the merged result back in the following clock cycle.

On the feature data side of the data table, the three main registers are $F_{C}$ for accumulating the feature vector for the current row, $F_{M}$ for reading DT[M] for merging the data, and $F_{P}$ for holding the data for the retiring node. With data forwarding, $F_{P2}$ is required as a shadow for $F_{P}$ because of the synchronous memory semantics.

There are a wide range of feature vector combinations depending on the window pattern and data forwarding due to hazards, the most complex of which occurs for the pattern 
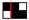
 when there is a link between the two runs on the previous row. This requires combining four feature vectors (requiring a minimum of three ⊕ operators): (1)$$F_{C}≔F_{C}⊕F\left(x,y\right)⊕F_{P2}⊕F_{M}.$$

The propagation delay for this combination is reduced by implementing these operations as a tree ($\left(F_{C}⊕F\left(x,y\right)\right)⊕\left(F_{P2}⊕F_{M}\right)$).

There are also four different destinations for the results: either of the two memory write ports, the current accumulation register $F_{C}$, or the data forwarding register $F_{P2}$ (when resolving hazards). The top ⊕ operator merges feature vectors to memory as they are retired. The bottom-left ⊕ operator accumulates the feature vector for object on the current row. The bottom-right ⊕ operator is used to merge the feature vector from the previous row into the current row. When a particular operation is not required, the null feature vector, $F_{0}$ is combined; this is an identity operation ($F⊕F_{0}=F$). Multiplexing a null feature vector (a constant) has a lower propagation delay than multiplexing after the ⊕ operation.

If the retired node has no links (M=0), then the final feature vector for the completed connected component is output from $F_{P}$ (or $F_{P2}$ if there was data forwarding).

## 5. Results and Discussion

For testing the design, the feature vector consisted of the bounding box and area of each connected component $\left\{x_{\min},x_{\max},y_{\min},y_{\max},area\right\}$. For this feature vector, the combination operator consists of the following:(2)$$F_{1}⊕F_{2}\equiv \left[\begin{array}{c} \min(x_{\min1},x_{\min2})\\ \max(x_{\max1},x_{\max2})\\ \min(y_{\min1},y_{\min2})\\ \max(y_{\max1},y_{\max2})\\ area_{1}+area_{2} \end{array}\right]$$

The design was compiled to hardware using Intel Quartus Prime 20.1.1 Lite Edition. It quickly became apparent that the design had long combinatorial delays. These were mitigated by pipelining the three main modules, with the data processing delayed by one clock cycle to enable the merge link from the link processing to be registered. The resources for the ‘balanced’ optimisation mode are shown in Table 1. Switching to the ‘performance’ optimisation mode gave a 3% improvement in maximum clock frequency, however, this was at the expense of a 17% increase in logic and 36% increase in registers required.

In terms of critical path, this design is fairly well balanced, with several paths having similar propagation delays. Within the data processing module, one critical path is from the data table output register, $F_{M}$, through the two feature vector combination operations (⊕) with the result written back to memory. This path is required to handle the pattern 
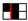
 when there is a link between the two runs on the previous row. The data for these are combined, with the result merged with the feature vector of the current row, and written to the data table ($DT\left[C_{W}\right]≔\left(F_{P2}⊕F_{M}\right)⊕F_{C}$). A similar critical path is for writing the result into the $F_{C}$ register ($F_{C}≔\left(F_{P2}⊕F_{M}\right)⊕(\left[F_{C}⊕\right]F\left(x,y\right)$, which occurs with the pattern 
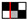
). In both cases, the majority of the propagation delay comes from the ⊕ operations. This means that these paths could potentially be pipelined by delaying the processing of the current row by one cycle relative to the processing of the previous row (although this would introduce additional timing complications).

However, the critical path within the link processing module has a similar propagation delay. It runs from the output of the link table ($Data_{R}$ in Figure 8) through the update and replace blocks, into the cache, $L_{P}$. While the update and replace blocks are primarily multiplexers, the critical path involves checking whether or not the link being added is already present (which is used to control the multiplexers). Adding a pipeline register within this path would delay the loading of $L_{P}$ by one clock cycle, potentially introducing additional memory conflicts and read-before-write hazards to those identified in Section 4.2.

In both cases, the processing is made more complex by the constraint not to introduce an additional clock cycle (containing a background pixel) at the end of each row. If this was allowed, the queuing requirements of the link processing reduce, and many of the read-before-write hazards within the data processing disappear. Both of these would reduce the logic required, and potentially result in a higher maximum clock frequency.

Overall, the clock frequency of 106.5 MHz is sufficient to process 1920×1080 images at 51.4 frames per second (assuming no blanking).

### 5.1. Comparison with Other CCA Algorithms

The primary goal of the Union-Retire algorithm is to reduce the memory requirements of implementing single-pass CCA on an FPGA. Therefore, the comparison will focus primarily on this aspect of the implementations. There are typically four memory-based data structures required to implement CCA on an FPGA.

The first of these is the data table, which contains the feature vectors of the partially completed components. The width of the data table depends only on the feature vector being extracted (extracting more features requires a wider data table). For single-pass CCA algorithms the depth of the data table depends on the number of active labels, which for current algorithms, corresponds to half of the width of the image. Consequently, the size of the data table is generally independent of the particular algorithm. In terms of data processing, a minimum of two ⊕ operators are required, one for accumulating the feature vector of the current pixel, and one for managing component mergers.

The second data structure manages the connectivity, mergers, or equivalences, between labels. This is typically a lookup table that maps a label to the current representative label. The width of the table is generally the width of a label, although may be wider if additional information must be stored [8].

The third data structure is a row buffer, which makes processed labels on one row available for processing the following row, and is a consequence of the window required for propagating labels and determining connectivity. The length of the row buffer generally corresponds to the width of the image (although it can be shorter if preprocessing is used to simplify the image contents by removing noise, and run-length encoding the relevant information [21]). The width of the row buffer is usually the width of a label (for propagating labels from one row to the next), although it can be reduced to a single bit for deterministic labelling [3,20].

Finally, there may be additional auxiliary data structures as required by the algorithm, for example for

Recycling labels;Keeping track of chains of successive mergers;Detecting completed components.

#### 5.1.1. Memory Requirements of Union-Retire

The requirements for the data table for UR-CCA are the same as for other algorithms (for extracting a given feature vector). The data structure for managing connectivity is the link table, which has the width of two labels (or indices in this paper). Since UR-CCA uses deterministic labelling, the row buffer is only a single bit wide. One advantage of the UR-CCA algorithm is that no additional memory-based auxiliary data structures are required.

Table 2 compares the resource requirements of several reported systems with the proposed Union-Retire-based architecture. Note that since reported results differ in terms of image size, extracted feature vectors, and the FPGA technology used, it is difficult to make a meaningful direct comparison from Table 2. However, the differences between the proposed architecture and listed architectures are identified and discussed in the following.

#### 5.1.2. Comparison to Single-Lookup CCA

The single-lookup algorithm [8] represents the most recent refinement of the original single-pass CCA algorithm (from [6]), where the representative label for a pixel in the previous row is found using a single lookup of the label from the row buffer.

Unlike the other methods compared here, single-lookup CCA requires processing time at the end of each row to manage chains of mergers. While for typical images, this processing time is small (typically less than 1% [6]), in the worst case it can average 20% overhead per line [1,6], with the pathological worst-case pattern giving 50% overhead on some lines [6]. Indeed, it was the goal of the other algorithms compared (including UR-CCA) to eliminate this variable factor from the processing [2].

To correctly select the representative label in the case of mergers, single-lookup CCA requires the label to be augmented with the row number [1,8]. This makes the merger table of similar width to the link table used by UR-CCA.

The row buffer makes the assigned labels available for processing the next row, requiring the row buffer to hold a row of labels. UR-CCA significantly reduces the width of the row buffer (to one bit) by using deterministic labelling.

The single-lookup CCA algorithm requires several auxiliary data structures. Two stacks are required to enable the single lookup operation: one to resolve chains of mergers at the end of each row, and one to manage the case where a single lookup is insufficient to give the representative label. A queue is used for recycling labels, enabling labels (and associated memory resources) to be reused. This is not required by UR-CCA, because the deterministic labelling allocates and retires labels in a sequential order enabling the queue to be replaced by a single counter. Finally a set of active tags are used by a parallel process to detect completed components. These are not required by UR-CCA because the retirement operation is able to directly detect completed components as they happen.

Consequently, UR-CCA requires significantly less memory for processing a given image size. Although the other resources are harder to compare (because of technology differences), UR-CCA does require more complex processing to maintain the link table structure, and for managing hazards with the data table processing. This highlights a trade-off between reducing the memory at the expense of an increase in logic resources. Part of the difference in clock speed can be attributed to the Cyclone series being lower cost than the higher-performing Kintex series.

Note that a more optimised double-lookup version has been described [1], although the associated FPGA resources have not been documented. While this reduces some of the auxiliary data structures, and the number of memory accesses required per pixel, it has many of the limitations of the single-lookup algorithm.

#### 5.1.3. Comparison to Zig-Zag Scan

To eliminate the end of row processing, Bailey and Klaiber [11] observed that the EOL resolution pass consists of a reverse scan along the row. Therefore, by using a zig-zag scan, merger chains could be unwound while processing the next line. Compared to the single-lookup CCA, this removes the need for the two stacks, although the other auxiliary data structures are still required, as is an additional one-bit row buffer to convert a regular raster scan into a zig-zag scan. The data table is also larger for the zig-zag scan because it needs to incorporate tags for detecting the end of completed components. The memory requirements for UR-CCA are consequently about 36% lower than for the zig-zag scan.

UR-CCA also requires fewer registers, but more logic because of the cost of managing data hazards. The more complex processing also gives a slightly reduced maximum clock frequency.

#### 5.1.4. Comparison to Direct Relabelling


Jeong et al. [10] eliminated the end of row overhead by directly updating all of the old labels in the row buffer with the new representative label whenever a merger occurs. This eliminates the need for a table to maintain the connectivity, but instead requires the row buffer to be content addressable (to enable the old labels to be found and replaced). On an FPGA, this requires the row buffer to be implemented in logic rather than using memory blocks, hence the large resource count and lower operating speed compared to UR-CCA.

The implementation also limited the processing to only 127 labels. While this may be adequate for small images, or with preprocessing to eliminate small noise components, it is unable to process all possible images. This reduces the RAM required for the data table, hence the lower RAM requirements than UR-CCA.

#### 5.1.5. Comparison to Linked-List Based Processing


Tang et al. [20] introduced the deterministic labelling method of allocating labels to runs of pixels sequentially within the image, reducing the row buffer to a single pixel wide. However, the algorithm described in [20] only processes four-connected components. Eight-connected components can be converted to four-connected components, although this requires an extra bit within the row buffer to enable correct accumulation of the feature vector.

The connectivity is represented using a linked list, with three pointers per label (one to point to the list head, which is the representative label; one to the next run in the set; and one to the list tail, where new runs are added). Mergers are managed by manipulation of the pointers within this list structure. UR-CCA improves on this by moving the focus away from finding a representative label, and focusing solely on the connectivity. This reduces the number of links required for each label from three to two, effectively reducing the size required by the link table by 33%. The cost of moving away from a linear structure to a more general graph representation is more complex processing. This is reflected in the increase in logic resources required by UR-CCA (although the Virtex and Cyclone have different logic cell architectures making direct comparison difficult). However, the trade-off between memory and logic resources is again apparent.

The feature vector reported in [20] consists only of the bounding box, which will require less memory for the data table. Measuring the area as well would further increase the memory from that reported in Table 2.

#### 5.1.6. Discussion of UR-CCA

Overall, the adoption of the Union-Retire paradigm results in a significant reduction in the memory required to implement single-pass CCA on an FPGA. Use of deterministic labelling of runs enables the row buffer to be reduced to a single bit width (a total of *W* bits for an image of width *W*). This cannot be reduced any further. Maintaining the connectivity through the link table requires only two pointers per node. This is accomplished by building chains within the connectivity graph. Occasionally, though, the restriction of only two links does require additional processing (when a link needs to be added to an already full node). The cost of this is a small price for the significant reduction in memory. The data table cannot be reduced, although this is common for all single-pass algorithms. No additional auxiliary memory structures are required for ensuring correct connectivity or for determining completed objects. Therefore, it is difficult to see how the memory requirements could be further reduced.

One bottleneck of the algorithm is the bandwidth required to access the link table. The architecture described here detects several pixel patterns that result in redundant operations. This detection logic significantly increases the complexity of the control logic. One potential avenue for improvement is to split the link table into two parallel banks, containing the even and odd indices in separate banks. This would double the available bandwidth with almost no additional memory cost, and may be able to simplify the control logic.

Much of the complexity of the data processing results from read-before-write hazards, which must be detected and then resolved using data forwarding. This is relatively expensive, especially for wide-feature vectors, because an additional multiplexer is required for each bit within the feature vector. Most of these hazards result from managing EOL conditions where there is not a background pixel between two successive object pixels. While this issue may be avoided by inserting a background pixel between each row, this would violate the constraint of no row overheads. However it may be more productive to relax this constraint (since the timing is fixed; it is always one pixel per row), and run the connected component analysis at a slightly higher clock rate than the incoming pixel stream (with appropriate clock domain synchronisation on the input).

Some of the complexity of the data table processing results from the decision to perform a ‘lazy’ retire, merging the data at the end of a run on the previous row. It may be worth exploring whether merging the feature vectors earlier can reduce the processing complexity. Potentially, this could enable the accumulation to be pipelined, reducing the propagation delay.

The depth of on-chip memories is a power of two. This can cause more memory to be used than strictly required for images where the width is a power of two (for example 1024×768) because both the link table and data table are of depth $\lfloor \frac{W}{2}\rfloor +3$ (from 0 to $\lfloor \frac{W}{2}\rfloor +2$). The next largest power of two almost doubles the memory allocated for the tables. The 0 index is required to indicate empty links, so strictly speaking, this does not require an entry within the tables. Index 0 can be used for holding links and data by making the observation that a node never links to itself. Therefore, self-links could be used to represent empty links. The other two extra indices are required to ensure that a node is retired before the index is allocated again on the following line. (This case occurs where every second pixel is an object pixel.) However, such links would always belong to the same connected component, so if an index is reallocated before it is retired, the retirement step can simply be skipped. These changes would enable the depth of the link and data tables to be reduced to $\left\lceil \frac{W}{2}\right\rceil$, although at the complexity of some additional control logic.

## 6. Conclusions

Although the Union-Retire algorithm of [3] is conceptually quite simple, an FPGA implementation poses several challenges. In software, each pixel can be processed completely before considering the next pixel. However, in hardware several clock cycles are required to read and update the memory structures representing the connectivity. To process one pixel per clock cycle, the memory accesses must be pipelined. For the link table, the main complexity is detecting and eliminating redundant memory operations to enable the memory bandwidth requirements to be met. For the data table, the main complexity is the deep processing path to manage mergers onto the current row, and to resolve read-before-write data hazards through data forwarding.

The dominant resource required by single-pass CCA algorithms is memory. When compared with other architectures that have no end-of-row processing overhead, it has been shown that the memory requirements are reduced by 36% compared to zig-zag based connected components analysis, and the link table is reduced by 33% compared to linked-list based processing. However this reduction in memory has come at the cost of a modest increase in logic resources required for pipelining, in particular managing memory bandwidth and overcoming data hazards. The significant memory reduction reduces the footprint on an FPGA, enabling a smaller device to be used. This makes Union-Retire-based processing the most memory-efficient design to date.

## Figures and Tables

**Figure 1 jimaging-08-00089-f001:**
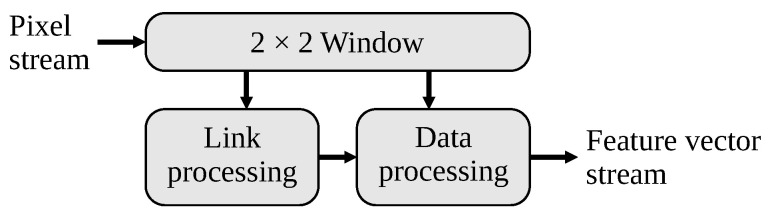
The basic hardware architecture for UR-CCA.

**Figure 2 jimaging-08-00089-f002:**
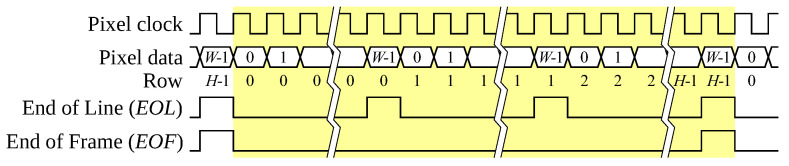
Continuous pixel stream, and associated control signals. The stream for one frame is highlighted.

**Figure 3 jimaging-08-00089-f003:**
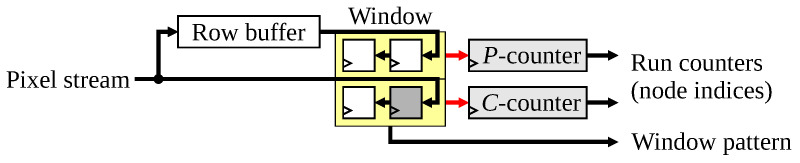
Architecture for constructing the 2×2 window and providing node indices for each run. The current pixel within the window is shaded.

**Figure 4 jimaging-08-00089-f004:**
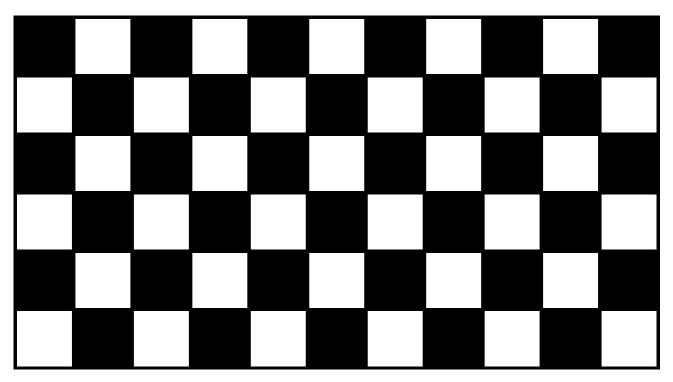
The worst case image pattern for raw memory bandwidth.

**Figure 5 jimaging-08-00089-f005:**
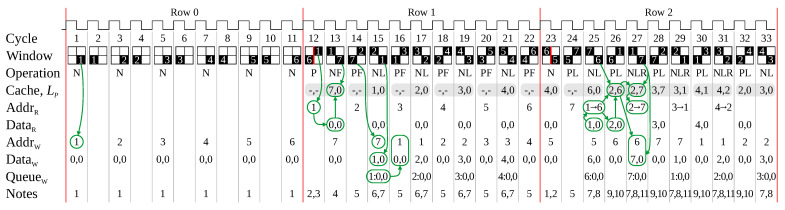
Memory accesses when processing the worst-case image of Figure 4. The red lines correspond to the EOL boundary. The run/node indices are shown for each 2×2 window. The -,- for the cache indicates that the cache content is invalid. In the read address for a link, the number after the → indicates the index to be added. Notes: **1**. The N-operation initialises LT[C] for the new node. **2**. There is no link from this pattern, because this is at the EOL and the start of the next. **3**. The P-operation reads LT[P] into the cache, $L_{P}$. **4**. The F-operation directly updates the cache, and no chain is required for the first link (rule 2). **5**. The P-operation takes priority over the L- or F-operation. As the cache has not been loaded, processing the link is deferred to the following cycle (rule 3), where. **6**. the cache is updated directly and the chained link can be written directly, without first reading (rule 4). **7**. The write for the N-operation is queued for the following clock cycle because the write port is busy (rule 9). **8**. The link directly updates the cache, and the chain is initiated. **9**. The link just read replaces the read for the P-operation, with the link write directly updated in the cache (rule 5). **10**. The current link does not need to be added because it is already in the cache (rule 6). **11**. The retired node has two links, requiring a link between them to be added.

**Figure 6 jimaging-08-00089-f006:**
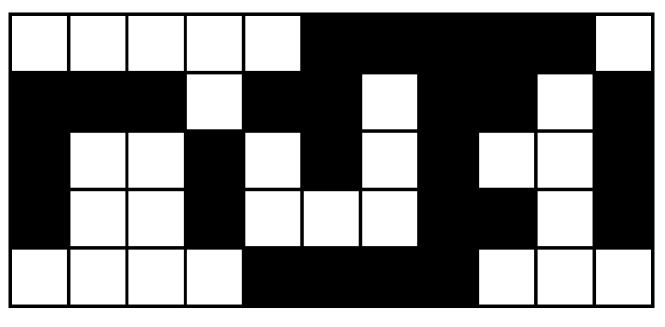
Another example image.

**Figure 7 jimaging-08-00089-f007:**
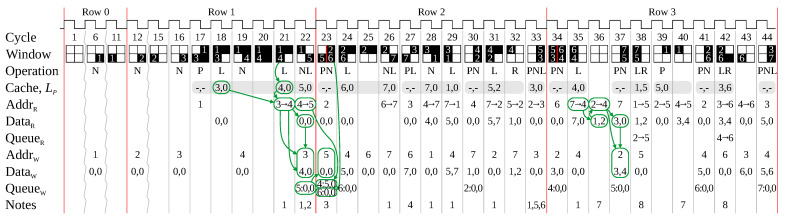
Memory accesses when processing the example image of Figure 6. Note that some cycles at the start (where nothing interesting is happening) are skipped. Notes: **1**. A chain operation updates the cache, and adds the chained link to memory. **2**. The N-operation is queued because the write port is busy (rule 9). **3**. Two additional writes are initiated (one from the previous chained link, and one for the new index), both are queued. **4**. The L-operation must be delayed because the cache has not been loaded yet (rule 3). **5**. The previous read is used to initialise the cache for the P-operation (rule 5). **6**. Since this is the last pixel in the row, an L-operation also started in this cycle. **7**. The node already has two links, so the link must be passed on (rule 7). **8**. Both an L- and R-operation are initiated this cycle; the retirement link is processed first and the chain queued (rule 8).

**Figure 8 jimaging-08-00089-f008:**
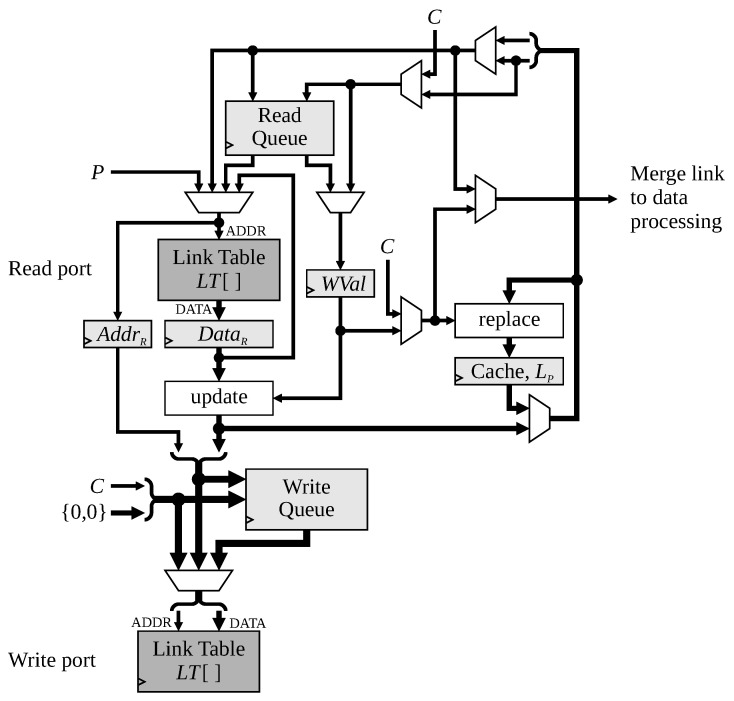
Detailed architecture of the link table processing. Only the data path is shown. The thickness of the lines represents the width of the data path (whether 1, 2, or 3 indices).

**Figure 9 jimaging-08-00089-f009:**
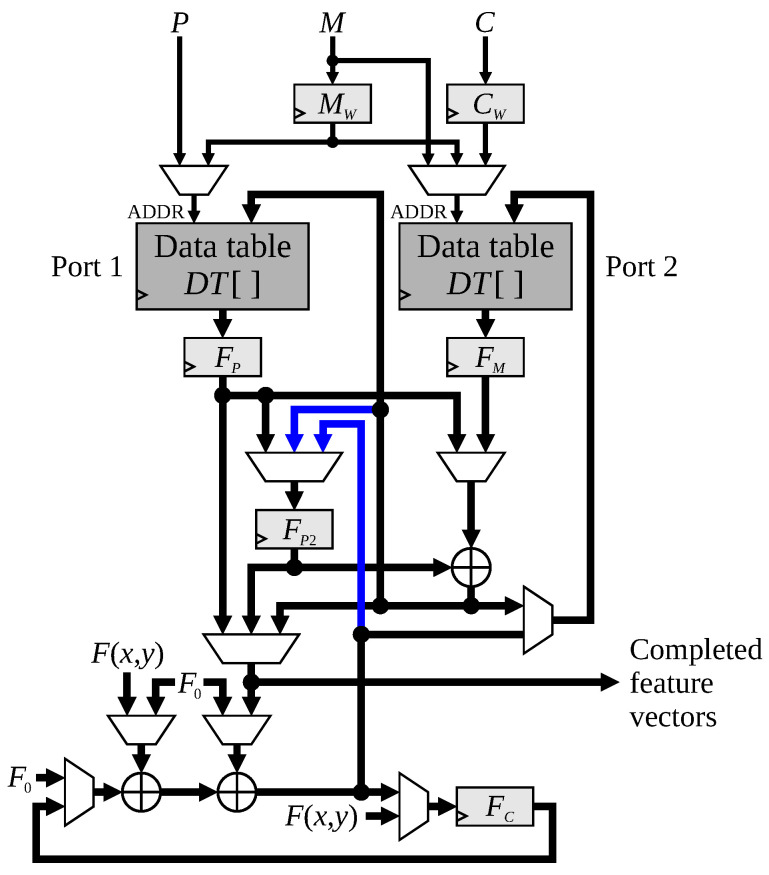
Detailed architecture of the data table processing. Only the data path is shown. The thicker lines represent the feature data. Data forwarding connections are shown in blue. $F_{0}$ represents the null feature vector (containing empty data).

**Table 1 jimaging-08-00089-t001:** Synthesis results using ‘balanced’ optimisation, for a 1920 × 1080 image, extracting the bounding box and area of each connected component. ALUTs are Intel’s adaptive lookup tables; FFs are the number of flip-flops or registers; M10K are the number of Intel’s 10 kbit RAM blocks.

Module	Intel Cyclone V 5SEMA5F31C6				
	ALUTs	FFs	RAM (bits)	M10K	$f_{\max}$
2×2 window	50	48	1920	1	
Link processing	243	137	19,260	2	
Data processing	839	349	62,595	7	
Total	1133	534	83,775	10	106.50 MHz

**Table 2 jimaging-08-00089-t002:** Comparison of several CCA hardware architectures. Abbreviations for the extracted feature vector are: (A) area, (FOM) first-order moment, (BB) bounding box.

Implementation of Architecture	Technology	Image Size (pixels)	Extracted Features	LUTs	Registers	RAM (kbits)	$f_{\max}$ (MHz)
Single lookup CCA	Kintex 7	256×256	BB	00,493	296	108	185.59
Klaiber et al. [8]		1920×1280		00,723	381	180	151.40
Direct relabelling ${}^{a}$	Cyclone IV	640×480	BB, FOM	36,478	N/A	018	060.58
Jeong et al. [10]		1920×1080		57,036	N/A	029	058.44
Linked-list	Virtex II	256×256	BB	00,543	187	072	104.26
Tang et al. [20]		640×480		00,654	227	092	097.07
	Cyclone V	640×480	BB, A	00,778	539	053	113.05
Zig-zag scan		1920×1080		00,906	587	131	114.48
Bailey and Klaiber [11]	Kintex 7	640×480	BB, A	00,907	499	092	185.49
		1920×1080		0,1343	564	166	192.16
This work	Cyclone V	640×480	BB, A	0,1013	490	025	109.97
Union-Retire CCA		1920×1080		0,1133	534	084	106.50

${}^{a}$ Hardware resources are for a maximum of 127 labels [10].

## Data Availability

No new data were created or analysed in this study. Data sharing is not applicable to this article.

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
