# Peer review of "Union-Retire for Connected Components Analysis on FPGA"

_2313-433X, 2022, doi:10.3390/jimaging8040089_

Round 1

Reviewer 1 Report

In the article, the authors presented the implementation of the Union-Retire algorithm in the FPGA system. They also showed a significant reduction in memory requirements and the link table.

At the end of the article, a list of abbreviations is provided, which significantly facilitates the analysis of the article. It is worth adding a similar table with the mathematical notations used in the presented equations.

At least 7 out of 22 cited articles are written by the authors. Of course, I understand the need to refer to previous works, but the number of self-citations in relation to the number of all cited publications is disturbingly high (over 30%). This suggests the lack of a proper analysis of the literature, which largely cites the works of the authors.

Author Response

In the article, the authors presented the implementation of the Union-Retire algorithm in the FPGA system. They also showed a significant reduction in memory requirements and the link table.

At the end of the article, a list of abbreviations is provided, which significantly facilitates the analysis of the article. It is worth adding a similar table with the mathematical notations used in the presented equations.

Thank you for your suggestion. We have added the combination operator, the M, P, and C indices, and F(x,y) into the list of abbreviations.

At least 7 out of 22 cited articles are written by the authors. Of course, I understand the need to refer to previous works, but the number of self-citations in relation to the number of all cited publications is disturbingly high (over 30%). This suggests the lack of a proper analysis of the literature, which largely cites the works of the authors.

The reason for the large number of self-citations is that we have been particularly active in this area over the last 15 years or so, and this is a relatively niche topic. If you look at the review papers ([1] and particularly [2]) you will see that we have given fair coverage to the range of relevant prior work within the topic. We have removed one of the self-references.

Reviewer 2 Report

This paper describes a hardware architecture based on the Union-Retire CCA  algorithm Authors have previously published. The manuscript is well written and the comparison is comprensive. However, the content of Table 2 is somewhat confusing and it seems that competitors perform much better than the proposed system. Even though Authors wrote that "it is difficult to
make a meaningful direct comparison from Table 2" and they give an explanation for each alternative system,  I suggest to improve the Table content substituting the LUTs numbers with the used number of bits, thus removing the technological dependency, and eventually adding power dissipation information.

Author Response

This paper describes a hardware architecture based on the Union-Retire CCA  algorithm Authors have previously published. The manuscript is well written and the comparison is comprensive. However, the content of Table 2 is somewhat confusing and it seems that competitors perform much better than the proposed system. Even though Authors wrote that "it is difficult to make a meaningful direct comparison from Table 2" and they give an explanation for each alternative system,  I suggest to improve the Table content substituting the LUTs numbers with the used number of bits, thus removing the technological dependency, and eventually adding power dissipation information.

Referring to the number of bits within the LUTs only partially helps, but even the 6-LUTs used by Intel and Xilinx have a different underlying architecture. That is why we have made a detailed discussion of the comparison, which we believe is a fair analysis of different implementations. There is a trade-off between logic resources, memory resources and speed. The focus of the UR-CCA algorithm is on reducing the memory, which is often a critical resource on smaller, low-cost FPGAs. The significant drop in memory required has come at the expense of a modest increase in logic resources. Sentences to this effect have been added within the comparison section and the conclusions to explicitly state this trade-off.

For many of the designs, power dissipation is either not available, or only an estimation provided by modelling, making it largely dependent on resources, fmax, and an often arbitrarily estimated toggle rate.

Reviewer 3 Report

It is an interesting work, well written, well structured and very well analyzed and compared with other hardware algorithms. 

In general, the hardware implementation of an algorithm is a complex task in the sense that a lot of time is invested in the design. This time is reflected in the economic cost and the ASIC implementation is highly dependent on the demand of the product (algorithm, application etc). 
It is true, that the design constraints and the demand of the design, conditions the use of a software based architecture or a hardware based architecture.

If the architecture proposed by the author is useful to develop it in embedded systems, as the author comments, my question is, 

 Has the author compared or implemented such architecture in any embedded system where the UR-CCA has been implemented software. It would be interesting, add to the article, explanations and / or details of this. 

In the same sense, I would like to know with data/results or that the author explains that the constraints, economic (high cost of hardware development so that an ASIC is economically justified) and design constraints such as memory resources used, processing times, power consumption, imposed by the application, justify the use of a hardware architecture for the implementation of the algorithm.

Author Response

It is an interesting work, well written, well structured and very well analyzed and compared with other hardware algorithms. 

In general, the hardware implementation of an algorithm is a complex task in the sense that a lot of time is invested in the design. This time is reflected in the economic cost and the ASIC implementation is highly dependent on the demand of the product (algorithm, application etc). It is true, that the design constraints and the demand of the design, conditions the use of a software based architecture or a hardware based architecture.

If the architecture proposed by the author is useful to develop it in embedded systems, as the author comments, my question is, 

 Has the author compared or implemented such architecture in any embedded system where the UR-CCA has been implemented software. It would be interesting, add to the article, explanations and / or details of this. 

In the same sense, I would like to know with data/results or that the author explains that the constraints, economic (high cost of hardware development so that an ASIC is economically justified) and design constraints such as memory resources used, processing times, power consumption, imposed by the application, justify the use of a hardware architecture for the implementation of the algorithm.

Thank you for your comments. You are right, the market is unlikely to ever be large enough to make an ASIC for this task to be economical. Therefore we have removed references to ASIC from the paper (abstract and conclusion). However, an FPGA implementation, which is relatively cost efficient even with low numbers, is relevant. We have added a paragraph at the start of the introduction to provide some of our motivation behind an improved FPGA implementation.

We have written a software implementation of Union-Retire CCA, but the purpose of this was to validate the algorithm rather than for comparison with hardware. I agree, the algorithm also has potential to make a software implementation of CCA more efficient, but that was beyond the scope of this paper.